

# Yearling proportion correlates with habitat structure in a boreal forest landbird community

Peter Pyle[1], Kenneth R. Foster[2], Christine M. Godwin[2], Danielle R. Kaschube[1] and James F. Saracco[1]

[1] The Institute for Bird Populations, Petaluma, CA, USA
[2] Owl Moon Environmental Inc., Fort McMurray, AB, Canada

## ABSTRACT

Landbird vital rates, such as productivity and adult survivorship, can be estimated by modeling mist-netting capture data. The proportion in which an adult breeding bird is 1 year of age (a "yearling"), however, has been studied only minimally in a few landbird species. Here we relate yearling proportion to habitat-structure covariates, including reclamation age, in a boreal forest landbird community. Data were collected at 35 constant-effort mist-netting stations over a 6-year period, and consisted of 12,714 captures of adults, of 29 landbird species, including 4,943 captures of yearlings. Accuracy of age determination (yearling or older) was assessed based on recapture data and error rates were estimated at a mean of 8.1% (range 0.0–19.4%) among the 29 species, with 20 species showing age-error rates <10%. The estimated mean yearling proportion was 0.407, ranging from 0.178 to 0.613 among species. Remote-sensed Enhanced Vegetation Index (EVI), a measure of habitat greenness, was positively correlated with age since reclamation up to 20 years, at which time it became comparable to that of natural stations. The probability of capturing a yearling for species associated with mature forest was lower at stations with higher EVI and the opposite was the case for species favoring successional habitats. These results suggest that yearling birds are being excluded from preferred breeding habitats by older birds through despotism and/or that yearlings are simply selecting poorer habitat due to lack of breeding experience or other factors. This dynamic appears to be operating in multiple species within this forest landbird community. Captured yearlings may also be "floaters", or non-breeding individuals not holding territories. However, presuming that yearlings show lower reproductive success whether floating or not, our results suggest that stations with high yearling proportions could be located within sink as opposed to source habitats. Overall, we infer that yearling proportion may become an important vital-rate measure of habitat quality and reclamation efforts, when combined with indices of population size, productivity, reproductive condition and survivorship.

Corresponding author
Peter Pyle, ppyle@birdpop.org

## INTRODUCTION

Vital rates such as productivity and survivorship have increasingly been used to assess the status of landbird populations and inform conservation actions (*DeSante, Nott & O'Grady, 2001*; *Anders & Marshall, 2005*; *Saracco, DeSante & Kaschube, 2008*; *Robinson, Julliard & Saracco, 2009*; *Rushing et al., 2015*). While trends in landbird abundance are useful for highlighting species of conservation concern, the assessment of vital rates may help identify causes of declining trends; for example, whether they are driven by factors on or away from breeding grounds (*Newton, 2004*; *Albert et al., 2016*). Data on vital rates collected at constant-effort landbird capture stations can also be used to predict population viability (*Ryu et al., 2016*) and can be modeled as functions of habitat variables to inform conservation-management strategies (*DeSante, 1995*; *Saracco et al., 2016*, *2018*), including reclamation and restoration programs (*Foster et al., 2017*). To date, modeled vital-rate terms have primarily included those of productivity, often indexed as the probability that a captured bird at a station had fledged that year and survivorship of breeding adults, as estimated from capture-mark-recapture models (*Saracco, DeSante & Kaschube, 2008*).

Population age structure within breeding populations of iteroparous species has received less attention but may reflect population dynamics and aspects of habitat quality (*Rodenhouse, Sherry & Holmes, 1997*). In landbirds, advances in distinguishing yearlings from older adults for many species (*Jenni & Winkler, 1994*; *Pyle, 1997*) provide new opportunities to estimate proportions of first-year breeding adult landbirds in populations. Yearling proportions have been found to correlate negatively with landbird population densities (*Graves, 1997*; *Sillett & Holmes, 2005*) and positively with marginal, lower-quality, or disturbed habitats, likely the result of competitive exclusion of yearlings by older birds from higher-quality breeding habitats (*Graves, 1997*; *Holmes, Marra & Sherry, 1996*; *Hunt, 1996*; *Bayne & Hobson, 2001*; *Haché & Villard, 2010*). Yearling proportion has also correlated positively with reproductive success within a population the previous year (*Sillett, Holmes & Sherry, 2000*), reflecting a direct demographic effect. Yearling proportions from capture stations have the potential to increase our understanding of landbird demography, habitat selection and annual recruitment rates; however, assessments of yearling proportion thus far have been limited to a few directed studies on individual species in which one-year-old landbirds, in most cases males only, are readily distinguished from older birds in the field.

Here we examine yearling proportions using data from 18,799 captures of 12,714 individual landbirds, of 29 species (Table 1), captured at 35 Monitoring Avian Productivity and Survivorship (MAPS) constant-effort bird-capture stations over a 6-year period in the oil sands region of northeastern Alberta (Fig. 1). The goals of this study were to (1) examine the accuracy and utility of yearling proportion as a measure of habitat structure in a boreal-forest landbird community, (2) assess the relationship between the probabilities of capturing a yearling and habitat based on multi-species hierarchical models and (3) assess locally measured and remote-sensed structural vegetation changes in relation to habitat reclamation maturity (e.g., number of years since reclamation

**Table 1 Yearling proportion correlates with habitat structure in a Boreal forest landbird community.** Sample summaries, age error rates and yearling proportions for landbird species captured in the boreal forest of northeastern Alberta. Four-letter alpha codes presented here are used in Figs. 4 and 5. Number of stations in which the species was captured is indicated (Stas.) Year-unique individuals (Year-Inds.) indicate the summed number of individuals captured per year (including between-year recaptures). Numbers of yearling adults (SY), older adults (ASY) and adults undetermined to age (AHY) are given for each species. Age error rates were calculated as the proportion of instances an age determination of SY or ASY was changed for recapture records, as verified by recapture data, among the given sample. Yearling proportion calculated as SY/(SY + ASY).

| Common name | Scientific name | Code | Stas. | Year-Inds. | SY | ASY | AHY | Age error rate ($n$) | Yearling prop. |
|---|---|---|---|---|---|---|---|---|---|
| Yellow-bellied sapsucker | *Sphyrapicus varius* | YBSA | 33 | 267 | 132 | 120 | 15 | 0.056 (178) | 0.524 |
| Alder flycatcher | *Empidonax alnorum* | ALFL | 36 | 845 | 166 | 573 | 106 | 0.119 (345) | 0.225 |
| Least flycatcher | *Empidonax minimus* | LEFL | 35 | 517 | 209 | 252 | 56 | 0.086 (163) | 0.453 |
| Red-eyed vireo | *Vireo olivaceus* | REVI | 31 | 507 | 78 | 360 | 69 | 0.042 (143) | 0.178 |
| Black-capped chickadee | *Poecile atricapillus* | BCCH | 29 | 190 | 48 | 62 | 80 | 0.071 (99) | 0.436 |
| Ruby-crowned kinglet | *Regulus calendula* | RCKI | 18 | 155 | 48 | 62 | 45 | 0.021 (47) | 0.436 |
| Swainson's thrush | *Catharus ustulatus* | SWTH | 36 | 747 | 276 | 356 | 115 | 0.046 (263) | 0.437 |
| American robin | *Turdus migratorius* | AMRO | 35 | 471 | 204 | 210 | 57 | 0.132 (151) | 0.493 |
| Cedar waxwing | *Bombycilla cedrorum* | CEDW | 27 | 358 | 142 | 191 | 25 | 0.017 (60) | 0.426 |
| Chipping sparrow | *Spizella passerina* | CHSP | 38 | 801 | 302 | 427 | 72 | 0.080 (224) | 0.414 |
| Clay-colored sparrow | *Spizella pallida* | CCSP | 23 | 755 | 299 | 399 | 57 | 0.128 (344) | 0.428 |
| Savannah sparrow | *Passerculus sandwichensis* | SAVS | 5 | 266 | 71 | 179 | 16 | 0.140 (136) | 0.284 |
| Song sparrow | *Melospiza melodia* | SOSP | 15 | 141 | 39 | 88 | 14 | 0.134 (97) | 0.307 |
| Lincoln's sparrow | *Melospiza lincolnii* | LISP | 35 | 546 | 203 | 277 | 66 | 0.180 (417) | 0.423 |
| Swamp sparrow | *Melospiza georgiana* | SWSP | 23 | 216 | 119 | 75 | 22 | 0.088 (159) | 0.613 |
| White-throated sparrow | *Zonotrichia albicollis* | WTSP | 37 | 1474 | 450 | 853 | 171 | 0.089 (1093) | 0.345 |
| Ovenbird | *Seiurus aurocapilla* | OVEN | 37 | 567 | 212 | 278 | 77 | 0.084 (178) | 0.433 |
| Northern waterthrush | *Parkesia noveboracensis* | NOWA | 19 | 127 | 19 | 88 | 20 | 0.061 (66) | 0.178 |
| Black-and-white warbler | *Mniotilta varia* | BAWW | 30 | 261 | 117 | 113 | 31 | 0.026 (77) | 0.509 |
| Tennessee warbler | *Oreothlypis peregrina* | TEWA | 37 | 2399 | 968 | 1148 | 283 | 0.055 (660) | 0.457 |
| Mourning warbler | *Geothlypis philadelphia* | MOWA | 20 | 220 | 73 | 130 | 17 | 0.077 (195) | 0.360 |
| Common yellowthroat | *Geothlypis trichas* | COYE | 28 | 236 | 67 | 145 | 24 | 0.081 (111) | 0.316 |
| American redstart | *Setophaga ruticilla* | AMRE | 22 | 214 | 85 | 106 | 23 | 0.049 (81) | 0.445 |
| Magnolia warbler | *Setophaga magnolia* | MAWA | 27 | 292 | 143 | 133 | 16 | 0.073 (219) | 0.518 |
| Yellow warbler | *Setophaga petechia* | YEWA | 21 | 309 | 106 | 167 | 36 | 0.041 (197) | 0.388 |
| Yellow-rumped warbler | *Setophaga coronata* | YRWA | 29 | 278 | 124 | 137 | 17 | 0.194 (93) | 0.475 |
| Canada warbler | *Cardellina canadensis* | CAWA | 22 | 304 | 129 | 121 | 54 | 0.133 (165) | 0.516 |
| Wilson's warbler | *Cardellina pusilla* | WIWA | 28 | 193 | 70 | 93 | 30 | 0.132 (91) | 0.429 |
| Rose-breasted grosbeak | *Pheucticus ludovicianus* | RBGR | 22 | 134 | 44 | 81 | 9 | 0.000 (33) | 0.352 |

commenced) as it may affect species-specific and habitat-specific variation in yearling proportions.

# MATERIALS AND METHODS

## MAPS stations

We established 35 landbird-capture MAPS stations in the oilsands region of northeastern Alberta, Canada (Fig. 1), to monitor landbird demographics. Stations were located in landscapes that included both riparian and upland habitats (*Foster et al., 2017*). Six MAPS

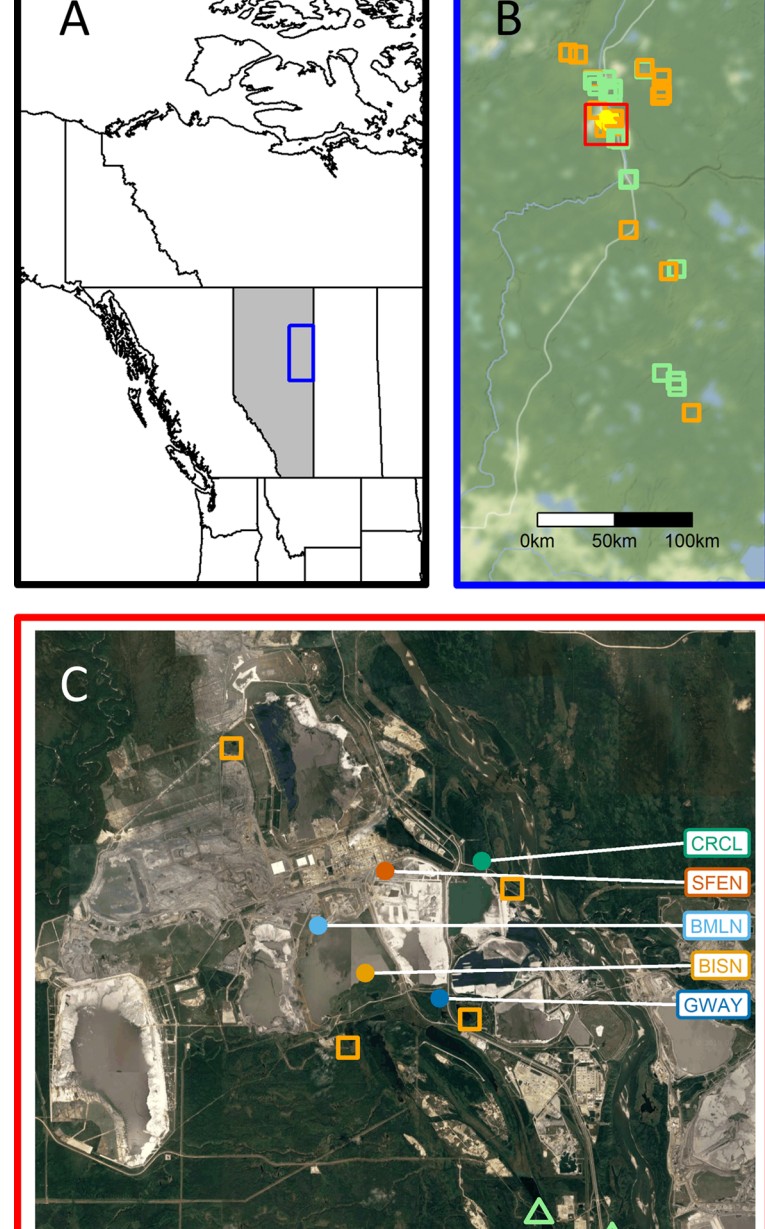

**Figure 1 Study region.** (A) Study region within Alberta, Canada. (B) MAPS station locations; stations comprised largely (>90%) of natural boreal habitats ($n = 15$) are iindicated with green squares, those affected by disturbance from resource development activities ($n = 15$; 0–89% natural habitats) with open orange squares and stations in reclaimed sites ($n = 5$) are indicated with open yellow circles. (C) Surrounding habitats for five reclaimed stations, with color coding corresponding to that in Fig. 2; see *Foster et al. (2017)* for station codes. Map data for (B) Stamen Design, under CC BY 3.0. Data by OpenStreetMap, under ODbL; Map data for (C) Google, TerraMetrics.

stations were operated in 2011, 24 stations were operated in 2012 and 30–35 stations were operated each year in 2013–2016; once a station was established it was operated in all following years, excepting a single station which did not operate in 2013 due to flooding and five stations that were inaccessible in 2016 due to a regional wildfire. Stations were operated once every 10-day period between 5 June and 7 August, for a total of 6 days of operation per station per year. On each day of operation, between eight and 14, fixed-location, 12 m mist-nets were operated for 6 h, beginning at local sunrise. Captured landbirds were fitted with uniquely numbered leg bands. Birds were aged following *Pyle (1997)* as hatching year (HY) or older (after-hatching-year; AHY), and most AHYs were further determined to be either one-year-old yearlings (second-calendar-year; SY) or older (ASY) adults. Adults undetermined to SY or ASY were aged AHY. Prior to each season, biologists were extensively trained on criteria to distinguish SY and ASY individuals of captured species. Time permitting, digital images were obtained of the body plumage, wings and tails of birds initially aged AHY that, following detailed examination of the images, resulted in age revisions to SY or ASY.

## Accuracy of age determinations

Our analyses were based on 13,790 year-unique adult captures (different individuals captured in a given year), including 12,714 individuals of 29 species (Table 1). Of the total number of captures, 4,943 were aged as yearling adults (SY), 7,224 were aged as adults older than yearlings (ASY) and 1,623 were of adults that could not be precisely aged (AHY), indicating that 88.2% (range 57.9% for black-capped chickadee to 94.5% for magnolia warbler; see Table 1 for scientific names) of AHY birds were aged to SY or ASY. Other than black-capped chickadee (57.9% aged) and ruby-crowned kinglet (71.0%), we aged >82% of AHYs to SY or ASY among the remaining 27 species (Table 1).

The ability to separate ASY from SY birds is critical to inferring yearling proportion, as errors may be biased toward either of these two age classifications, skewing results. Of 18,799 total adult captures (including within-season captures), 4,754 (25.3%) were initially aged AHY and 3,131 of these (65.9%) were re-determined to be SY or ASY through analysis of digital images from one or more captures. Many of the remaining AHY birds (Table 1) were either processed during times of high capture volume and were released without obtaining photographs, or were of early-molting species (e.g., black-capped chickadee) that had completed or nearly completed the prebasic molt and could not be aged to SY or ASY by the previous year's flight feathers (*Pyle, 1997*). The total number of AHY birds in the data set, 1,623, represents 11.8% of our sample.

Recaptured birds were aged independently of previous age determination; thus, data on birds captured on multiple occasions allowed us to infer error rates. The mean percent of captures in which age was re-assessed (from SY to ASY or vise versa) according to verified recapture data was 8.1%, ranging from 0.0% of 32 recaptured rose-breasted grosbeaks to 19.4% of 93 recaptured yellow-rumped warblers (Table 1). Twenty of our 29 species showed age-error rates <10% (Table 1).

## Vegetation data

We collected in-situ, local habitat data on 1–5 broad habitat types at each station within 100 m of each station's periphery (*Foster et al., 2017*). Habitat types were delineated based on plant community composition, vegetation structure and hydrology. For each station and habitat type we estimated cover within three vegetation strata: (1) understory cover (0.5–5 m), (2) midstory canopy cover (5–15 m), and (3) upperstory canopy cover (>15 m). Previously we have also assessed the proportions of each station that consisted of "natural", "disturbed" and "reclaimed" habitats, with the proportion of natural habitats (excluding open water) ranging from 0% to 98% (*Foster et al., 2017*). For the present analysis, 15 of the 35 stations, those with >90% of the habitat undisturbed, were considered to be "natural" stations. For the five stations in which >55% of the habitat was reclaimed ("reclaimed" stations), the age since reclamation, calculated as the difference between year of initial vegetation restoration (e.g., tree planting) and year of data collection, ranged from 1 to 34 years. The remaining 15 stations, which included natural vegetation cover of between 50% and 88%, were classified as "disturbed" stations. To derive station-scale habitat metrics, we calculated weighted averages of each of these variables, with weights equal to the estimated proportion of each habitat type present at the station.

For remote-sensed habitat data we used 16 day, 0.25 km resolution, Enhanced Vegetation Index (EVI) data derived from the Moderate Resolution Imaging Spectroradiometer instrument of NASA's Terra satellite (MODIS product MOD13Q1; http://terra.nasa.gov/). We examined relationships between vegetation greenness and: (a) in-situ habitat parameters, (b) % natural cover and (c) probability of capturing a landbird yearling at each station. The EVI is a composite metric which incorporates structural and seasonal components of habitat quality including primary productivity (leaf chlorophyl content), leaf area, canopy cover and vegetation complexity (*Glenn et al., 2008*), and which has previously been correlated with landbird occurrence and vital rates (*Saracco et al., 2016*). We extracted EVI cell values using the MODIS Subsets function of the MODIS Tools package 30 (*Tuck et al., 2014*) in the statistical software package R (*R Core Team, 2015*) at the 0.25 × 0.25 km scale for 81 cells surrounding each MAPS station, a grid extending 1.13 km in cardinal directions from station centers and a projected area (5.11 km$^2$) estimated as being sampled by a MAPS station (*DeSante & Kaschube, 2009*). For each station, we averaged values for the 81 cells across two June dates to obtain a single EVI value at each station for each year from 2000 to 2016 (2000 is the first year for which MODIS data are available).

## Data analysis

To investigate habitat changes at reclaimed stations we created plots of cover percentages (for the three vegetation layers collected in situ) against years since reclamation at the five reclaimed stations and for reference, the 15 natural stations. To provide insights into which vegetation layers contributed most strongly to variation in EVI, we plotted EVI values for the five reclaimed stations against years since reclamation for each year between 2000 and 2016 (a total of 85 data points). We also used a two-variable linear

regression with year- and site-specific June EVI as the response variable and percent natural cover and years since reclamation as explanatory variables. We included both the five reclaimed stations and 15 disturbed stations in this analysis, considering un-reclaimed stations as being zero years since reclamation. The model was fit in R (*R Core Team, 2015*) and bivariate relationship plotted using the "scatterplot3D" function in the R package scatterplot3D (*Ligges & Mächler, 2003*). A quantile plot of standardized residuals and residuals v. fitted values plot suggested adequate model fit.

Our analyses of yearling proportion included 29 landbird species with a mean of ≥20 individual adult (SY, ASY and AHY) birds captured per year over the 6-year period (Table 1). To investigate the relationship between yearling proportion and EVI we implemented a multispecies hierarchical model using data from all 29 target species. We assumed the age of year-unique individuals to be Bernoulli random variables with success probability $p_{i,j,k,t}$, whereby the $i$, $j$, $k$, and $t$ represent indices for individual, species, station and year, respectively. Thus, $p_{i,j,k,t}$ indicated the probability that a year-unique capture of an adult individual represented an SY bird. We modeled $p_{i,j,k,t}$ with a logit-linear model:

$$\text{logit}(p_{i,j,k,t}) = \alpha_{j[i]} + \beta_{j[i]} \times \text{evi}_{k,t}$$

whereby $\alpha_{j[i]}$ represented random species intercepts and $\beta_{j[i]}$ the species-specific regression coefficients for the relationship with the June EVI value at station $k$ in year $t$, $\text{evi}_{k,t}$. We modeled $\alpha_{j[i]}$ and $\beta_{j[i]}$ as normally distributed with mean $\mu_a$ and $\mu_\beta$, and variances $\sigma_a$, and $\sigma_\beta$. We centered the continuous covariate, $\text{evi}_{k,t}$ around zero prior to analysis to facilitate estimation. We implemented the model using Bayesian methods in JAGS (*Plummer, 2003*) from R (*R Core Team, 2015*) using the R package "jagsUI" (*Kellner, 2015*). See Supplemental Information for details and model code. For this analysis, AHY individuals were included as unknown-age adults, and thus were estimated based on the model constraints and priors.

### Data availability and ethics statement

Our study was conducted in accordance with North American Banding Council (https://www.nabanding.net/) and MAPS (http://www.birdpop.org/docs/misc/MAPSManual18.pdf) protocols which minimize the impact of netting and processing on the health and safety of captured landbirds. All birds were captured and banded following protocols and permits issued by the Canadian Wildlife Service Bird Banding Office (Master Permit 10858) and the Alberta Government. The datasets generated and/or analyzed during the current study are available from the Harvard dataverse: https://doi.org/10.7910/DVN/NP2P2V.

### RESULTS

Extent of understory cover as measured in situ showed little relationship with station age since reclamation (Fig. 2A) whereas extent of midstory canopy cover increased with age since reclamation and was similar to midstory cover of natural stations by 30 years post reclamation (Fig. 2B). Extent of upperstory canopy cover among reclaimed stations

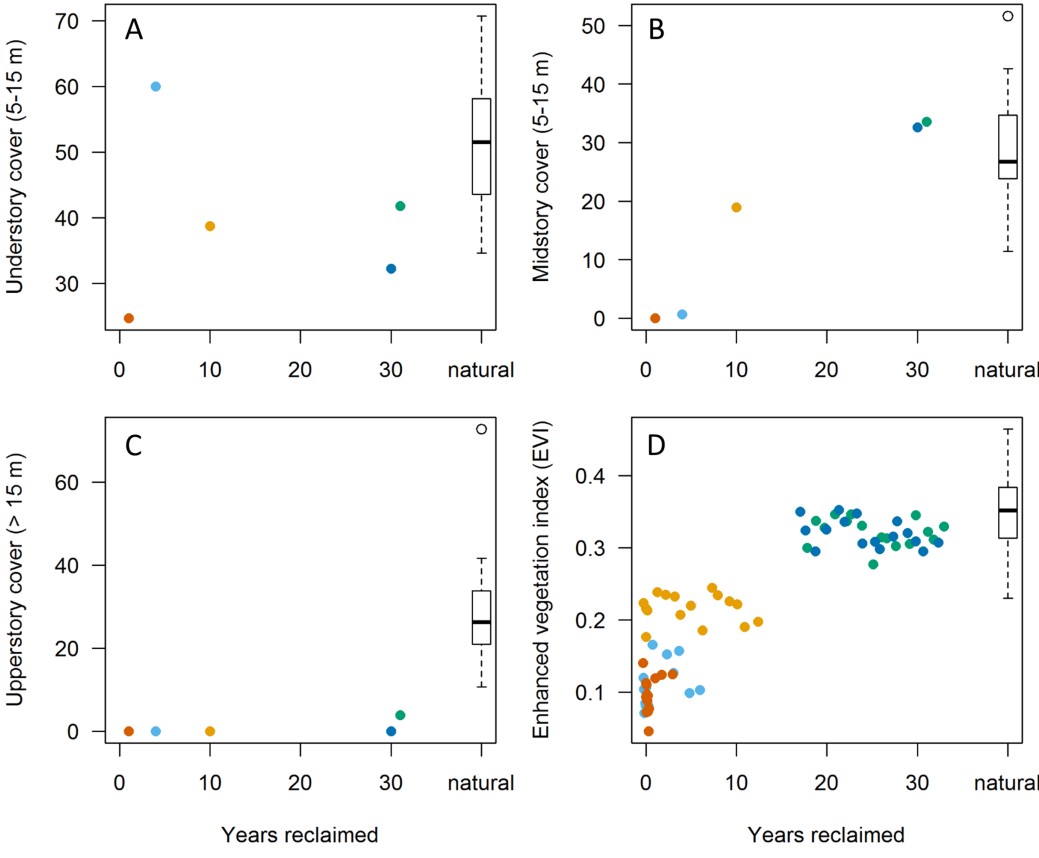

**Figure 2 Relationships between habitat variables and years since initiation of reclamation at 15 MAPS stations dominated by natural vegetation and five stations with predominantly reclaimed habitat.** Colors denote individual reclaimed stations (see Fig. 1). Natural sites (those with >90% natural boreal habitats; $n = 15$) are aggregated into boxplot to right. (A–C) Points represent values from in-situ habitat assessments in 2013. (D) Points represent June Enhanced Vegetation Index (EVI) values from up to 16 years per station (among years 2000–2015). Note that both understory and upperstory cover showed little correlation with years since reclamation whereas mid-story cover showed substantial increases, approaching those of natural sites by 30 years post reclamation. EVI values appear to reflect mid-story development, equating with natural stations at 15–20 years post reclamation.

also showed little relationship with station age up to 30 years, as compared to values derived from matured natural stations (Fig. 2C). Remote-sensed, EVI values increased with years since reclamation, with the index equating to those of natural stations up to about 20 years since reclamation (Fig. 2D). When we compared values among the five reclaimed stations and the remaining 30 stations, the EVI index also correlated with both year since reclamation and proportion of a station's habitat considered natural (Fig. 3).

The mean yearling proportion of the 29 species over all 6 years and 35 stations combined was 0.407, ranging from 0.178 in red-eyed vireo and northern waterthrush to 0.613 in swamp sparrow (Table 1). The probability of a captured adult being a yearling was negatively related to EVI (Fig. 4A), indicating that yearlings overall were found in less-forested habitats. This relationship was strongest for birds typical of forested habitats, as reflected by our capture data of adults and weakest for species inhabiting earlier

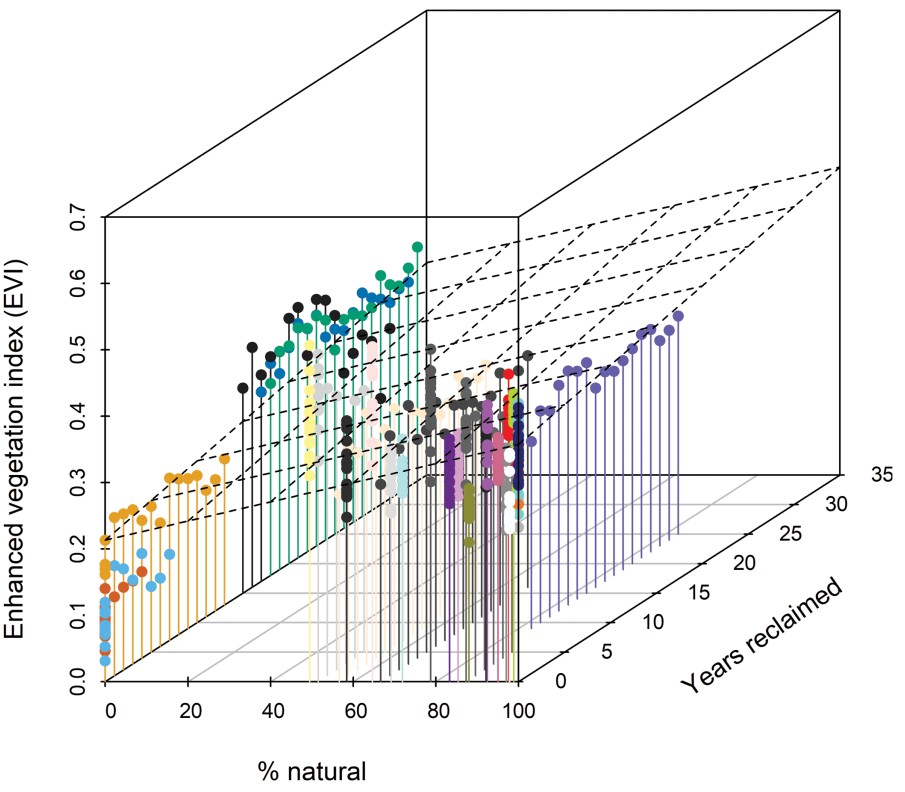

**Figure 3 Three-dimensional scatterplot indicating that Enhanced Vegetation Index (EVI) increases with % natural vegetation and years since habitat reclamation commenced.** The dashed plane shows the bivariate linear regression fit ($R^2 = 0.35$). Standardized regression coefficient for % natural was 0.064 (95% CI [055, 0.074]) and for years reclaimed was 0.030 (0.021, 0.040). Colors distinguish individual stations (see Figs. 1 and 2); other colors represent the 15 stations affected by disturbance from resource development activities, ranging from 0% to 89% of the station estimated to be in natural habitats (*Foster et al., 2017*).

successional habitats (Figs. 4B and 4C). The probabilities of capturing yearlings for species preferring mature forests, such as magnolia warbler, cedar waxwing, northern waterthrush and mourning warbler, were lower in those forested habitats and higher in successional habitats, whereas species preferring successional habitats, such as song sparrow, clay-colored sparrow, alder flycatcher, savannah sparrow and chipping sparrow, had lower yearling probabilities in those habitats and higher probabilities in forested habitats (Figs. 4B and 5). We also found significant relationships between EVI coefficients, EVI mean and EVI range among our 29 species, including a slightly positive correlation between EVI coefficient and range (Fig. 5), suggesting that more specialist species, those found in a narrower range of habitats, tended to have more negative correlations between EVI and yearling probabilities than species found in a wider range of habitats.

## DISCUSSION

Among our 35 stations, habitat-development patterns indicate that ground and understory cover do not change markedly after initial planting, that midstory cover develops within the first 30 years after initiation of reclamation and that uppperstory canopy does not

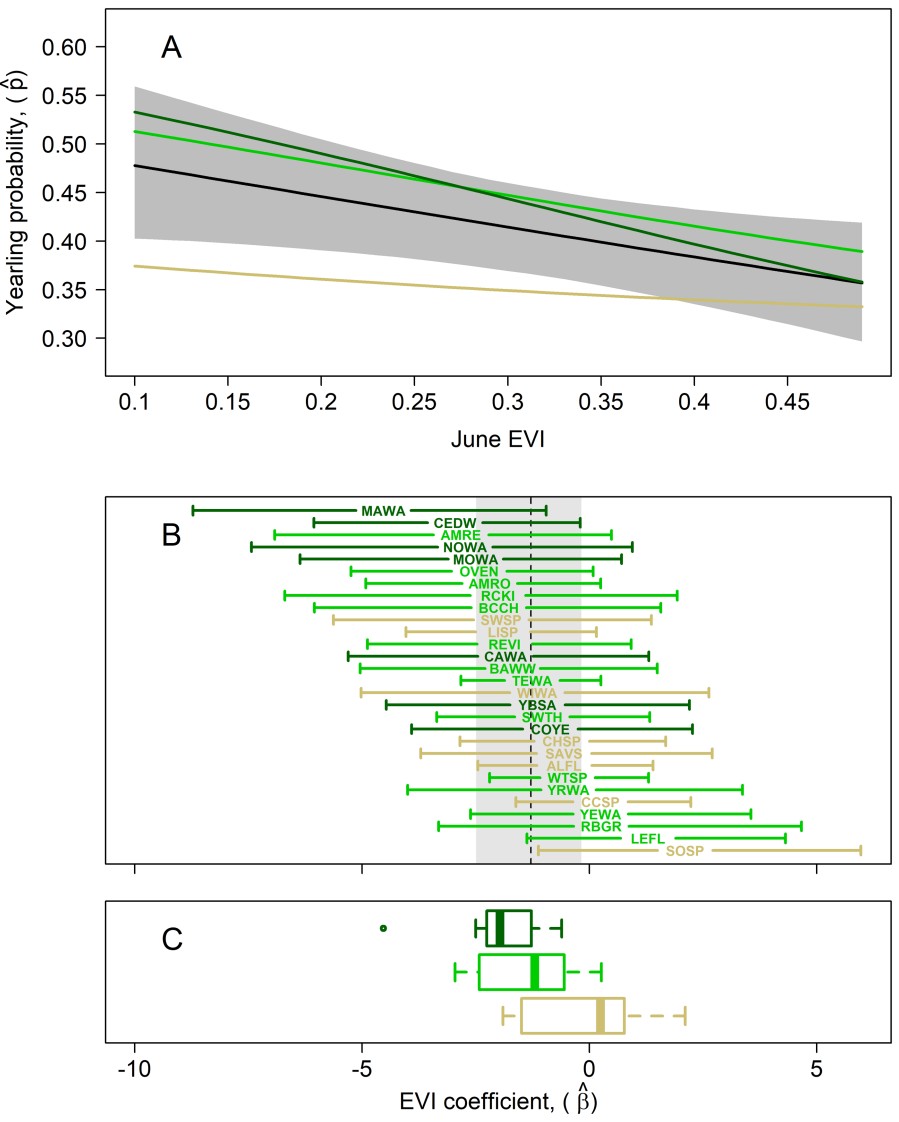

**Figure 4 Results of multispecies hierarchical model to investigate the relationship between the age structure of adult birds and Enhanced Vegetation Index (EVI).** Mean estimates for species in which adults were captured at highest rates in successional (tan; lowest EVI quartile), intermediate (light green; middle two EVI quartiles), and mature forest (dark green; upper EVI quartile) habitats, as measured by our in-situ habitat structure data, are shown to highlight potential habitat-related differences in yearling responses. (A) Estimated mean yearling probability (black line) ± 95% cred. int. (gray polygon) across observed range of EVI values ($\mu_\beta = -1.29$; 95% cred. int.: $-2.49$, $-0.18$). (B) EVI coefficient estimates (± 95% cred. int.) for each of the 29 target species, sorted from most negative (top) to most positive (bottom) EVI coefficient; species codes are defined in Table 1. The estimated mean species-EVI effect is indicated by a dashed line and gray region delineates the 95% credible region. (C) Box plots show distribution of mean coefficients for the 3 species groupings. The mean EVI coefficient for species with EVI values in the lowest quartile across species was $-0.55$ (95% cred. int.: $-3.64$, $3.67$), that for species in the middle half was $-1.38$ (95% cred. int.: $-5.60$, $2.88$), and that for species in the highest quartile $-1.95$ (95% cred. int.: $-6.72$, $1.83$). This indicates that yearling proportions for forest-dwelling species are lower in more-forested (greener) habitats whereas those of successional-habitat species is higher in forested habitats.

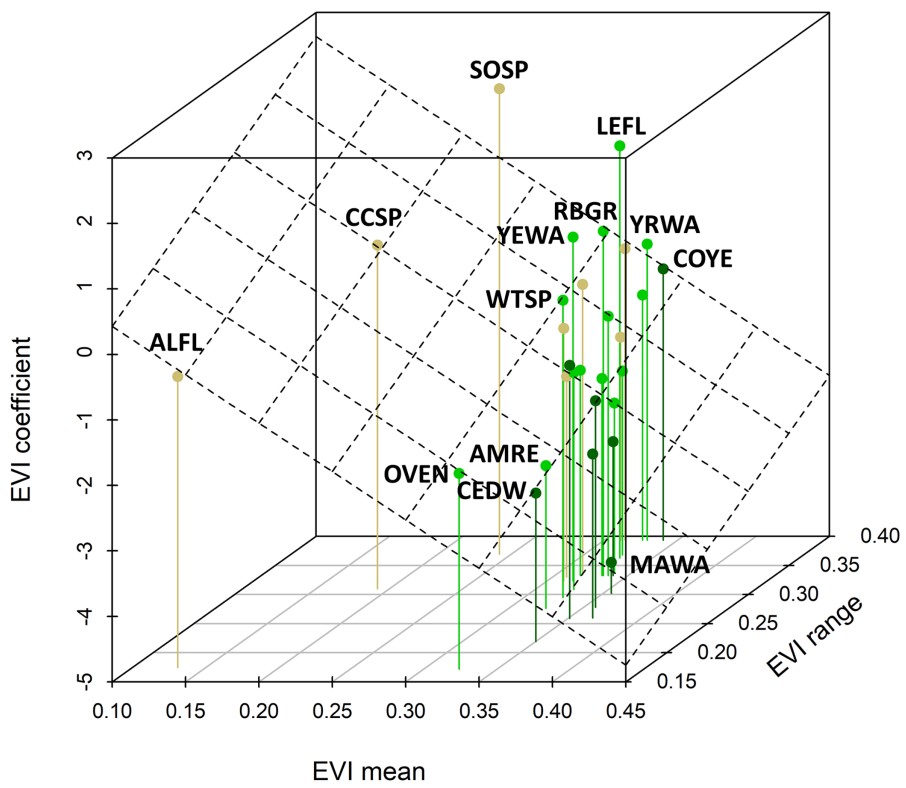

**Figure 5 Relationship between species coefficients as a function of the EVI mean and EVI range observed for that species.** The plane shows the bivariate regression fit ($R$-squared = 0.28). This relationship is highly significant (standardized regression coefficient = −0.717, 95% CI [−1.206, −0.278]) and there was a slightly positive relationship between the EVI coefficient and the EVI range. Four-letter codes (see Table 1) for outlying species are shown; see Fig. 4 for species-habitat color representation.

noticeably develop until >30 years after initiation of reclamation activities. This is consistent with other studies on boreal-forest habitat succession, in which trees reach about 40% of their maximum height within 50 years and attain maximum height and canopy development (closure) at about 75 years of age (*Bartels et al., 2016*). EVI values will increase as midstory canopy develops within the first 20 years, but then may not change substantially as upperstory canopies mature and obscure the lower strata. It thus appears that EVI can track habitat characteristics, structure and development for 20–30 years following vegetation planting, but that in-situ habitat data or data based on remote imagery that can distinguish canopy layers will be needed once upperstory canopy closure commences.

Mean yearling proportions among the 29 species examined show wide variation, from 0.178 in red-eyed vireo and northern waterthrush to 0.613 in swamp sparrow. We believe this variation reflects biological factors as opposed to age-determination or capture bias; following increased precision of age determinations though examination of digital images, mean error rate in age-determination was estimated at only 8.1%, with 20 of our 29 species showing age-error rates <10%. The variation in mean yearling proportions may reflect factors related to species-specific variation in (1) the ability or proclivity to
exclude yearling birds from higher quality habitats by adults, (2) age-specific variation in capture probabilities, and/or (3) habitat variables among our 35 MAPS stations.

Despite this variation in yearling proportions, forest-dwelling landbird species (cf. *Rodewald, 2015*), those captured at higher rates in forested stations of our study, showed lower probabilities of yearling capture in stations with higher EVI, whereas species captured more frequently in successional breeding habitats showed higher capture probabilities of yearlings in forested habitats. Both of these results are consistent with hypotheses suggesting that yearling birds are excluded through despotism to suboptimal breeding habitats (*Holmes, Marra & Sherry, 1996*; *Bayne & Hobson, 2001*; *Rohwer, 2004*; *Haché & Villard, 2010*). Alternatively, yearlings may also be selecting poorer habitats due to lack of breeding experience or other factors. In either case, it is possible that stations with high yearling proportions may occur in sink as opposed to source habitats (*Pulliam, 1988*; *Donovan et al., 1995*; *Faaborg et al., 2010*), presuming that yearling landbirds also average poorer reproductive success than older birds (*Holmes, Marra & Sherry, 1996*). Our study also indicates that despotic exclusion or other factors affecting yearling dynamics may operate selectively and specifically among different species within a forest landbird community. Maintenance of high yearling proportions in suboptimal habitats for the species may promote emigration of yearlings to more suitable habitats for their second year of breeding (*Greenwood & Harvey, 1982*; *Rohwer, 2004*), underscoring the need to model population age structure and habitat dynamics at the meta-population level over multiple years (*Faaborg et al., 2010*).

Using capture data from the same study areas reported on here, *Foster et al. (2017)* found significant correlations between habitat covariates and captures of adult birds, young (HY) birds and/or the probability of capturing a young bird (productivity). They also found that positive responses to reclamation age from obligate forest-dwelling species took more years to become evident than those for species preferring early successional-stage habitats. Our analyses further support relationships between demographic variables and habitat structure, including those related to habitat succession, suggesting that recruitment into the breeding population by yearling landbirds could be an initial indication of successful boreal-forest reclamation. We infer that yearling proportion can become an important vital-rate measure of both species-specific habitat suitability and the progress of reclamation efforts.

Captured yearlings may be "floaters", or non-breeding individuals not holding territories that are simply passing through the territories of breeding individuals (*Sherry & Holmes, 1989*; *Barber & Robertson, 1999*; *Bayne & Hobson, 2001*). Demographic analyses using capture data, for example those examining correlations of yearling proportion with population density or productivity (*Graves, 1997*; *Sillett, Holmes & Sherry, 2000*; *Bayne & Hobson, 2001*; *Rohwer, 2004*; *Haché & Villard, 2010*), should account for the possible presence of floaters. We suggest that breeding condition data, for example, the extent of brood-patch or cloacal-protuberance development (*Pyle, 1997*) or recapture data confirming length of stay, may be useful in assessing the occurrence of yearling and older floaters in capture data sets (*Barber & Robertson, 1999*). Habitat selection processes by age may also be subject to interannual or density-dependent

effects (*Rodenhouse, Sherry & Holmes, 1997*). By incorporating all demographic and breeding-condition variables in population models, that in turn include multiple species and multiple years of data, we suggest that yearling proportion may eventually be useful in estimating first-year recruitment and survivorship, the latter widely considered to be much lower than survivorship of older adults but is difficult to estimate due to low natal fidelity in most landbird species (*Hobson, Wassenaar & Bayne, 2004*; *Anders & Marshall, 2005*; *Cooper, Daniels & Walters, 2008*; *Faaborg et al., 2010*). In any case, presuming that yearlings show lower reproductive success than older birds, our results indicate that habitats with high yearling proportions could be located within sink as opposed to source habitats.

## CONCLUSIONS

To our knowledge, this study represents the first attempt to model the probability of capturing yearling landbirds against habitat factors using a multi-species approach. For many landbird species, age can only be accurately determined in the hand, so our dataset provides the only large-scale, long-term data set that we are aware of that permits estimation of yearling age-structure of both sexes combined. Key to deriving accurate yearling proportions is observer training in age-determination criteria and validation through a procedure like the photographic review and aging confirmation processes used here.

Our results suggest that yearling landbirds are excluded from optimal breeding habitats by older adults, although they might disperse into more optimal habitats the year following breeding as yearlings. The probability of capturing a yearling of forest-dwelling species was higher in successional-stage habitats and vise versa. In conjunction with other vital rates estimated using MAPS data (i.e., productivity, survivorship), yearling proportion may help identify sink habitats and estimate juvenile survival, on a species-by-species basis.

## ACKNOWLEDGEMENTS

We are greatly indebted to the many banders who helped collect data over the six-year period. P. Lai and L. Lade (OMEI) and R. Taylor, L. Helton, C. Ray, and R. Siegel (IBP) provided MAPS protocol-related expertise and program support. This is Contribution No. 590 of The Institute for Bird Populations.

### Funding

Funding in support of this project has been provided by Syncrude Canada Ltd., Hammerstone Corporation, Canadian Natural Resources Limited, Cenovus Energy, ConocoPhillips Canada Resources Corp., Devon Energy, Husky Oil Operations Ltd., Imperial Oil Ltd., Suncor Energy Inc., TOTAL E&P Canada, CNOOC International and the Oil Sands Developers Group. The funders had no role in study design, data collection and analysis, decision to publish, or preparation of the manuscript.

## Grant Disclosures

The following grant information was disclosed by the authors:
Syncrude Canada Ltd.
Hammerstone Corporation.
Canadian Natural Resources Limited.
Cenovus Energy.
ConocoPhillips Canada Resources Corp.
Devon Energy.
Husky Oil Operations Ltd.
Imperial Oil Ltd.
Suncor Energy Inc.
TOTAL E&P Canada.
CNOOC International.
Oil Sands Developers Group.

## Competing Interests

Peter Pyle, Danielle R. Kaschube, and James F. Saracco work for the Institute for Bird Populations. Kenneth R. Foster and Christine M. Godwin work for Owl Moon Environmental Inc. All of these authors declare that they have no competing interests.

## Author Contributions

- Peter Pyle conceived and designed the experiments, performed the experiments, prepared figures and/or tables, authored or reviewed drafts of the paper, and approved the final draft.
- Kenneth R. Foster performed the experiments, authored or reviewed drafts of the paper, helped secure funding, and approved the final draft.
- Christine M. Godwin performed the experiments, authored or reviewed drafts of the paper, helped secure funding, and approved the final draft.
- Danielle R. Kaschube analyzed the data, authored or reviewed drafts of the paper, and approved the final draft.
- James F. Saracco analyzed the data, prepared figures and/or tables, authored or reviewed drafts of the paper, and approved the final draft.

## Animal Ethics

The following information was supplied relating to ethical approvals (i.e., approving body and any reference numbers):

   Our study was conducted in accordance with North American Banding Council (https://www.nabanding.net/) and MAPS (http://www.birdpop.org/docs/misc/MAPSManual18.pdf) protocols which minimize the impact of netting and processing on the health and safety of captured landbirds. All birds were captured and banded following protocols and permits issued by the Canadian Wildlife Service Bird Banding Office (Master Permit 10858) and the Alberta Government.
## Data Availability

Data are available from:

Saracco, James, 2019, Replication data for "Yearling proportion correlates with habitat structure in a Boreal forest landbird community"

DOI 10.7910/DVN/NP2P2V,

Harvard Dataverse, V1, UNF:6:5hwOwqY1JeSjL7WfSQpuRA== [fileUNF]

DOI 10.7910/DVN/NP2P2V.

## Supplemental Information

Supplemental information for this article can be found online at http://dx.doi.org/10.7717/peerj.8898#supplemental-information.

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
