# Peer review of "Yearling proportion correlates with habitat structure in a boreal forest landbird community"

_PeerJ, doi:10.7717/peerj.8898_

## Round 0.1 · original submission · Major Revisions

The paper has now been reviewed by two expert reviewers. They agreed that this is an interesting and well-written manuscript. However, the reviewers also pointed out a number of points which have to be addressed or clarified, all of which I agree with.

I would like to ask you to carefully implement as many of the points raised as possible, and provide a detailed point-by-point reply to each of the reviewers' comments. I look forward to receiving a revised version of the manuscript!

Reviewer 1 ·

Basic reporting

na

Experimental design

na

Validity of the findings

na

Additional comments

This manuscript point out a potential interesting development to correlate yearling proportion from constant-effort landbird capture stations with habitat structures and which potentially can be used as a proxy for habitat quality. The manuscript is well written but I have some critical suggestion for improvement. Overall the manuscript lack ecological knowledge, this is necessary to be able to understand the species specific responses. Additionally a more thorough deviation of species into groups regarding their habitat preferences or functional groups is needed. This was done in figure 4 but this is unclear and should also be included in the methods section.

The use of proportion and probability combined is confusing throughout the manuscript. In my mind probability tells you the likelihood of something happening (model predictions), while proportion is just the comparison of (measurable) quantities, thus they do not tell the same and should not be mixed. In line, the title and research question are linked to yearling proportion but your models and figure 4 representing probabilities (predictions from the model). This results in the fact that you relate the probability that you can catch a yearling with habitat structure and not the proportion. Please clarify this throughout the manuscript. Find below some general comments:

Comments:

Line. 74. What about including sex? Is there a potential effect of sex on yearling proportion, this may differ between species, but competition and habitat exclusion may act different on male and females.

Line. 75. A little bit more information regarding species is required. Potential correlations between yearling proportions and habitat structure depend on habitat requirements of the studied species. More general ecology should be included and additionally it may be useful to define functional groups or groups representing their habitat requirements.

Line 79-80. I my opinion the second goal cannot be achieved within the current study setup. Habitat quality was not measured by a more reliable/known proxy and thus you cannot examine the accuracy and utility of yearling proportion as a measure of habitat quality. Just habitat (EVI, Greenness) not enough to classify high or low quality habitat, more precise measurements of for example reproduction is needed.

Line 98. Operating the stations until 7 August seems quite long. I am not completely familiar with the breeding season in this area but in general in July the nestlings fledged and birds start to move around more. This mean that your bias in capturing local breeding birds increase (higher influx from non-breeders and possible individuals from other areas after nest failure). Do you for example get the same result if you only use the data from June? Or can you correct for this effect in the models?

Line 174. Linear regression assume normal distribution of the data, even for time series. Thus in this case, could you provide extra information regarding this, I expect that data with N=5 cannot be normally distributed. If so, linear regression cannot be used, you can consider to use non-parametric correlation (Pearson correlation) instead.

Line 185-187. Why do you calculate probabilities as your research question is about proportion? I see you report probabilities in the result section L. 222-223, but you discus proportions in the discussion L. 260-263.

Line 222-230. Grouping the species in habitat classes is excellent but this is not written anywhere else in the intro or method. It would be good to know which species belong to which group and how they ecologically are divided.

Line. 238-259. Arrange according to the research question in the end of the introduction.

Line 248. Although habitat quality is a widely used concept, it can be misleading and vague. I think that within this study habitat quality is poorly defined and used in the wrong context. Within this study 29 species are used, all with differences in niche occupation. I think it is to straight forwards to say that EVI can track habitat quality, this is in my opinion species specific. Therefore this should be formulated in the way as the title indicate “Yearling proportion correlates with habitat structure” which is not similar as habitat quality. However, I do agree that yearling proportion can be an indication of habitat quality but this need more support.

Page 24, Figure 4. Be clearer about the species groups.

Reviewer 2 ·

Basic reporting

The paper “Yearling proportion correlates with habitat structure in a Boreal forest landbird community” reports bird banding data, specifically the proportion of yearlings, and its relationship to habitat factors in landscapes ranging from natural boreal forest to reclaimed land of different ages.
The paper is very well written and interesting. As far as I can judge, the statistical analyses are sound. I have only a few comments requiring some clarification:
L137: What was the spatial scale on which habitat was classified in situ?
L161: What was the rationale behind using exactly this spatial scale? Have the authors also evaluated other spatial scales or can they give reasons/references why this scale was considered?
L214: It is not entirely clear what is meant by "showed little development". Please rephrase and clarify
L238: It is unclear to which finding this first sentence is referring to. If this is just a general staement, consider removing it. I guess there are even some birds breeding in land which has not been reclaimed yet? Some ground breeding species? So, this statement makes - probably - only sense when referring to forest species

Experimental design

no comment

Validity of the findings

no comment

---

## Round 0.2 · accepted · Accept

Thank you very much for your revised re-submission. The reviewers were very happy with your revision of the manuscript and so am I! I just ask you to implement the minor point raised by one of the reviewers during the editing process. I look forward to seeing your paper online soon!

Reviewer 1 ·

Basic reporting

na

Experimental design

na

Validity of the findings

na

Additional comments

Dear authors,

Many thanks for adjusting the manuscript according to the suggested improvements. All the points of concern are adjusted to satisfaction and thus the manuscript is ready for publication.

There is one small detail: Reference in line 58 (Saracco, DeSante, & Kaschube, 2009) should be checked! In line 58 year of publication is changed to 2009, while in reference list (line 418-419) indicate 2008.

Kind Regards,

Reviewer 2 ·

Basic reporting

My previous commets were only minor and have been all well addressed

Experimental design

no comment

Validity of the findings

no comment